# Disparate Roles of Oxidative Stress in Rostral Ventrolateral Medulla in Age-Dependent Susceptibility to Hypertension Induced by Systemic l-NAME Treatment in Rats

**DOI:** 10.3390/biomedicines10092232

**Published:** 2022-09-08

**Authors:** Yung-Mei Chao, Hana Rauchová, Julie Y. H. Chan

**Affiliations:** 1Institute for Translational Research in Biomedicine, Kaohsiung Chang Gung Memorial Hospital, Kaohsiung 833, Taiwan; 2Institute of Physiology, Czech Academy of Sciences, 14200 Prague, Czech Republic

**Keywords:** nitric oxide, l-NAME, hypertension, rostral ventrolateral medulla, aging, reactive oxygen species, NADPH oxidase activator 1, mitochondria

## Abstract

This study aims to investigate whether tissue oxidative stress in the rostral ventrolateral medulla (RVLM), where sympathetic premotor neurons reside, plays an active role in age-dependent susceptibility to hypertension in response to nitric oxide (NO) deficiency induced by systemic l-NAME treatment, and to decipher the underlying molecular mechanisms. Systolic blood pressure (SBP) and heart rate (HR) in conscious rats were recorded, along with measurements of plasma and RVLM level of NO and reactive oxygen species (ROS), and expression of mRNA and protein involved in ROS production and clearance, in both young and adult rats subjected to intraperitoneal (i.p.) infusion of l-NAME. Pharmacological treatments were administered by oral gavage or intracisternal infusion. Gene silencing of target mRNA was made by bilateral microinjection into RVLM of lentivirus that encodes a short hairpin RNA (shRNA) to knock down gene expression of NADPH oxidase activator 1 (*Noxa1*). We found that i.p. infusion of l-NAME resulted in increases in SBP, sympathetic neurogenic vasomotor activity, and plasma norepinephrine levels in an age-dependent manner. Systemic l-NAME also evoked oxidative stress in RVLM of adult, but not young rats, accompanied by augmented enzyme activity of NADPH oxidase and reduced mitochondrial electron transport enzyme activities. Treatment with L-arginine via oral gavage or infusion into the cistern magna (i.c.), but not i.c. tempol or mitoQ_10_, significantly offset the l-NAME-induced hypertension in young rats. On the other hand, all treatments appreciably reduced l-NAME-induced hypertension in adult rats. The mRNA microarray analysis revealed that four genes involved in ROS production and clearance were differentially expressed in RVLM in an age-related manner. Of them, *Noxa1*, and *GPx2* were upregulated and *Duox2* and *Ucp3* were downregulated. Systemic l-NAME treatment caused greater upregulation of *Noxa1*, but not *Ucp3*, mRNA expression in RVLM of adult rats. Gene silencing of *Noxa1* in RVLM effectively alleviated oxidative stress and protected adult rats against l-NAME-induced hypertension. These data together suggest that hypertension induced by systemic l-NAME treatment in young rats is mediated primarily by NO deficiency that occurs both in vascular smooth muscle cells and RVLM. On the other hand, enhanced augmentation of oxidative stress in RVLM may contribute to the heightened susceptibility of adult rats to hypertension induced by systemic l-NAME treatment.

## 1. Introduction

Hypertension, a chronic condition of elevation in blood pressure (BP), is a well-known risk factor for the development of cardiovascular diseases (CVDs), including heart failure, coronary heart diseases, myocardial fibrosis, and infarction [1], atherosclerosis [1,2], stroke [1,3], and kidney failure [4]. The pathogenesis of hypertension is multifaceted, and one of the well-characterized causes is endothelial dysfunction resulting from nitric oxide (NO) deficiency [5,6,7]. Circulatory NO, derived primarily from endothelial NO synthase (eNOS) in vascular endothelial cells [8], participates in BP regulation via maintenance of vascular tone through the classical NO/soluble guanyl cyclase/cyclic guanosine monophosphate signaling to decrease intracellular calcium in vascular smooth muscle cells and evoke vasodilatation [9,10]. In addition, NO regulates cardiovascular functions by exhibiting anti-inflammatory [11], anti-platelet aggregatory [12], and anti-proliferative [13] actions. Accordingly, deficiency in NO production and/or availability leads to impaired vascular relaxation, increased vascular resistance, augmented platelet aggregation, and vascular inflammation or proliferation; all of which have been demonstrated to underpin the pathophysiology of hypertension [6,14,15].

A rat model of hypertension induced by the systemic administration of N^ω^-Nitro-l-arginine methyl ester hydrochloride (l-NAME), a NOS inhibitor, is widely used to mimic hypertension in human [16]. This treatment downregulates NOS expression in blood vessels [17], and reduces plasma NO levels [18], resulting in systemic vasoconstriction, increased vascular resistance, and hypertension [17,18,19]. l-NAME further aggravates the development of hypertension via antagonization of the anti-inflammatory, anti-platelet aggregatory, and/or anti-proliferative actions of NO [20,21]. In addition to its peripheral actions, l-NAME reportedly crosses the blood–brain barrier (BBB) to decrease NOS expression and inhibit NOS activity in brain [22,23,24], leading to brain-initiated hypertension. The neural mechanism(s) that underlies the pathogenesis of this form of hypertension induced by systemic l-NAME treatment is not fully understood.

The rostral ventrolateral medulla (RVLM), where sympathetic premotor neurons reside [25], is one of the major sites of action in the brain stem where NO participates in the neural control of cardiovascular functions [6,26,27,28]. Emerging evidence from preclinical studies in animal models suggests that NO in RVLM exhibits a predominantly sympathoinhibitory action in the regulation of sympathetic nerve activity (SNA) under physiological conditions [28,29]. Deficiency in tissue NO availability under various cardiovascular conditions, including hypertension [29], chronic heart failure [30], and metabolic syndrome [31,32], may lead to an increase in SNA because of the blunted sympathoinhibitory action of NO in RVLM. Moreover, NO participates in the neural control of cardiovascular systems via interactions with reactive oxygen species (ROS), in particular superoxide anion (O_2_^•^^−^) [6,26,32]. Under physiological conditions, NO is generated from its substrate L-arginine through coupling of NOS [33]. This coupling capacity is reduced under hypertensive conditions, wherein NOS removes an electron from its cofactor, NADPH, and donates it to an oxygen molecule to generate O_2_^•^^−^ rather than NO [32,33,34]. The causal role of ROS at RVLM in systemic l-NAME treatment-induced hypertension, nonetheless, has not been fully elucidated.

Epidemiological studies indicate aging is a predominant risk factor for CVDs. The prevalence of hypertension increases markedly with aging, attributed primarily to alterations in the structure, responsiveness, function, and rigidity of vessel walls [35], as well as dysregulation of the autonomic nervous system [36]. Several theories have so far been proposed to explain the etiology of biological aging [37]; among them, tissue oxidative stress was postulated to be a common denominator [38]. Indeed, a wide array of studies suggests the engagement of ROS in age-related CVDs, including hypertension, atherosclerosis, atrial fibrillation, and stroke [38]. Despite evidence suggesting causal links between oxidative stress, aging, and CVDs, there is a paucity in the literature on the molecular mechanisms that underlie age-related dysregulation of redox homeostasis and its contribution to the development of CVDs.

Redox signaling is critically involved in cardiovascular pathophysiology, and oxidative stress in RVLM has been demonstrated unambiguously to play a pivotal role in neural mechanisms of hypertension via the increase in neurogenic sympathetic outflow [6,14,26,27,28]. However, there is no current information on whether redox homeostasis in RVLM is affected by systemic l-NAME treatment. Furthermore, exactly how aging changes the redox state in RVLM, leading to its engagement in hypertension induced by systemic l-NAME treatment is unknown. This study therefore aims to investigate whether tissue oxidative stress in RVLM plays an active role in age-dependent susceptibility to hypertension in response to systemic l-NAME treatment, and to decipher the underlying molecular mechanisms. Our findings suggest that redox homeostasis in RVLM is tolerated in young (8 weeks of age) normotensive rats subjected to systemic l-NAME treatment; and NO deficiency, both in the vascular smooth muscle cells and RVLM, may likely be the key underlying molecular mechanisms. When animals turn adult (20 weeks of age), tissue ROS level in RVLM is augmented, leading to an enhanced increase in central sympathetic outflow that exacerbates hypertension induced by l-NAME.

## 2. Materials and Methods

### 2.1. Animals

Experiments were carried out in young (4 weeks old, *n* = 51) and adult (12 weeks old, *n* = 63) male normotensive Wistar-Kyoto (WKY) rats purchased from BioLASCO, Taipei, Taiwan. They were housed in animal rooms under temperature control (24 ± 0.5 °C) and 12 h light/dark (5 am to 5 pm) cycle. Standard laboratory rat chow (PMI Nutrition International, Brentwood, MO, USA) and tap water were available ad libitum. All animals were allowed to acclimatize for 14 days (young group) or housed to the age of 20 weeks (adult group) in an AAALAC-International accredited animal holding facility in Kaohsiung Chang Gung Memorial Hospital, Taiwan, before experimental manipulations. All experiments were performed in accordance with the guidelines for animal experimentation approved by our institutional animal care and use committee (no. 2018051701) as adopted and promulgated by the U.S. National Institutes of Health.

### 2.2. Implantation of Osmotic Minipump

Implantation of osmotic minipump into the peritoneal cavity or the cisterna magna was carried out according to previously reported procedures [32]. Briefly, for intraperitoneal (i.p.) implantation, animals were anesthetized with pentobarbital sodium (50 mg/kg, i.p.) and a midline incision of the abdominal cavity was made, followed by implantation of an Alzet^®^ osmotic minipump (model 2002, DURECT Co., Cupertino, CA, USA) into the peritoneal cavity. The incision was closed with layered sutures. Some animals also received implantation of an additional micro-osmotic minipump into the cistern magna. For intracisternal (i.c.) implantation, we performed a midline incision of the dorsal neck, and dissected the underneath muscle layers to expose the dura mater between the foramen magnum and C1 lamina. The dura mater was perforated with a 22-gauge steel needle, followed by insertion of a polyvinylchloride tubing into the cistern magna using an Alzet^®^ brain infusion kit. Observation of presence of cerebrospinal fluid at the outer end of the catheter assured the patency of the implantation. Tissue glue was used to seal the catheter to the dura mater and the incision was closed with layered sutures. A micro-osmotic minipump (Alzet^®^ 1007D) was positioned subcutaneously in the neck, and was connected to the outer end of the catheter. No obvious neurological signs were observed after minipump implantation. All animals received postoperative intramuscular injection of procaine penicillin (1000 IU). Only animals that showed recovery and progressive weight gain after the operation were used in subsequent experiments.

### 2.3. Blood Pressure and Heart Rate Measurement

Systolic blood pressure (SBP) and heart rate (HR) were determined in conscious animals with a tail-cuff sphygmomanometer (MK-2000; Momuroki Kikai Co., Tokyo, Japan). Baseline values were measured on days 3 and 1 prior to, on the day (day 0) before osmotic pump implantation, and then on days 1, 3, 5, 7, 9, 11, and 14 following i.p. l-NAME (10 mg/kg/day; Sigma-Aldrich, MA, USA) or 0.9% saline (vehicle control) treatment. During the recording sessions (at 14:00–16:00), rats were placed in restraint holders and the tail warmed on a warming pad for 10–15 min to increase blood flow and improve data acquisition. A full recording session consisted of 5 acclimatization cycles to optimize data acquisition, followed by 5 data acquisition cycles during which SBP and HR were recorded. The mean SBP and HR calculated for each time point were used for statistical analyses. It should be noted that SBP obtained by tail-cuff plethysmography in conscious rats has been validated to be comparable to those measured by radiotelemetry [32].

### 2.4. Evaluation of Sympathetic Vasomotor Activity

Sympathetic vasomotor activity was measured at the end of i.p. l-NAME infusion in young and adult animals from blood pressure recorded from a cannulated femoral artery in animals anesthetized via an anesthesia mask with isoflurane (5% for induction and 2% for maintenance). Each recording session lasted 30 min, and took place between 14:00 and 16:00. Temporal fluctuations in the low-frequency (LF, 0.25–0.8 Hz) component of the SBP signals were detected by continuous online, real-time spectral analysis based on fast Fourier transform using an arterial blood pressure analyzer (APR31a, Notocord, Le Pecq, France). The power density of the LF band was used as our experimental index to reflect sympathetic vasomotor tone [32].

### 2.5. Measurement of Plasma Norepinephrine

We measured plasma norepinephrine (NE) level by the o-phthalaldehyde (OPA) method using high-performance liquid chromatography (HPLC) with fluorescence detection. Plasma sample was mixed with ice-cold trichloroacetic acid, and centrifuged at room temperature. The supernatant was mixed with 4-fold methanol after being filtered through a syringe filter (0.22 µm; Chroma Technology Corp., Bellows Falls, VT, USA), centrifuged again at room temperature, and kept at −80 °C until analyses. NE was measured by HPLC as previously described [39], and comparing the area under the curve of each sample against standard NE solutions of known concentrations was used to compute the concentration (µg/µL). Each sample was analyzed in triplicates and the mean was used for statistical analyses.

### 2.6. Measurement of Serum Nitric Oxide

Serum NO concentration was assessed indirectly by measuring the levels of nitrate and nitrite, the stable end-product of NO, with a NO colorimetric assay kit (Arbor assay kit, Ann Arbor, MI, USA) according to manufacturer’s instructions. The absorbance of the solution was read on a microplate reader at 540 nm (ThermoFisher Scientific Inc., Waltham, MA, USA). Each sample was analyzed in triplicates and the mean was used for statistical analyses.

### 2.7. Measurement of Plasma Malondialdehyde

Assays for lipid peroxidation are commonly used for estimation of oxidative status. Levels of lipid peroxidation in plasma were measured by a malondialdehyde (MDA; a primary indicator of lipid peroxidation) assay kit (Biovision, Milpitas, CA, USA), following the protocol provided by the manufacturer. Briefly, plasma samples were reacted with thiobarbituric acid (TBA) at 95 °C for 60 min. A microplate spectrophotometer (ThermoFisher Scientific) was used to determine the level of MDA-TBA adduct with colorimetric absorbance read at 532 nm. Each sample was analyzed in triplicates and the mean was used for statistical analyses.

### 2.8. Measurement of Plasma Proinflammatory Cytokines

The levels of proinflammatory cytokines, including interleukin-1 β (IL-1β), IL-6, and tumor necrosis factor-alpha (TNF-α), in plasma, were measured using anti-rat ELISA Kits (ThermoFisher Scientific) according to the manufacturer’s specification. Plasma was collected and centrifuged for 10 min at 4 °C. The supernatants were used immediately to measure the concentrations of proinflammatory cytokines. Positive and negative controls were included on each plate. The final concentration of the cytokines was calculated by converting the optical density readings against a standard curve. Each sample was analyzed in triplicate and the mean was used for statistical analyses.

### 2.9. Tissue Collection and Protein Extraction from Rostral Ventrolateral Medulla

Animals were deeply anesthetized at the end of the experiments with an overdose of pentobarbital sodium (100 mg/kg, i.p.), followed by intracardial infusion with 500 mL of warm (37 °C) normal saline. The brain stem was rapidly removed and immediately frozen on ice. Using a rodent brain matrix (World Precision Instruments, Sarasota, FL, USA) and based on the atlas of Watson and Paxinos [40], the medulla oblongata covering RVLM was blocked between 0.5 and 1.5 mm rostral to the obex. Tissue from bilateral RVLM was collected by micropunches, and was stored at −80 °C until use.

For total protein extraction from RVLM, tissue micropunches were homogenized using a Dounce grinder with a tight pestle in ice-cold lysis buffer mixed with a cocktail of protease inhibitors (Sigma-Aldrich, St. Louis, MO, USA) to prevent protein degradation. Solubilized proteins were centrifuged at 20,000× *g* at 4 °C for 15 min, and the total protein in supernatant was quantified by the Bradford assay with a protein assay kit (Bio-Rad, Hercules, CA, USA).

### 2.10. Measurement of Reactive Oxygen Species in RVLM

RVLM tissues were homogenized in sodium phosphate buffer (20 mM), centrifuged and the supernatant was collected for ROS measurement by electron paramagnetic resonance (EPR) spin trapping technique, as described previously [41]. EPR spectra were captured using a Brucker EMXplus spectrometer (Bruker, Ettlingen, Germany). Typical parameters were set at microwave power: 20 mW, modulation frequency: 100 kHz; modulation amplitude: 2 G; time constant: 655.36 ms; conversion time: 656 ms; sweep time: 335.87 s. We added a membrane-permeable superoxide dismutase (SOD; 350 U/mL) into the incubation medium to determine ROS specificity. Spectra represented the average of 6 scans. Each sample was analyzed in triplicate and the mean was used for statistical analyses.

### 2.11. Measurement of Nitric Oxide in RVLM

For measurement of NO in RVLM tissue, tissue micropunches were homogenized in lysis buffer, centrifuged, and the supernatant was stored at −80 °C until use after being deproteinized using a Centricon-30 filtrator (Microcon YM-30, Bedford, MA, USA). Tissue NO levels were determined based on chemiluminescence reaction with the purge system of a NO analyzer (Sievers NOA 280^TM^, Boulder, CO, USA) [32]. Each sample was analyzed in triplicate and the mean was used for statistical analyses.

### 2.12. Measurement of Nitric Oxide Synthase Activity in RVLM

Tissue NOS activity in RVLM was measured according to previously reported procedures [32]. RVLM tissues were lysed and centrifuged to obtain the supernatants, which were then used for detection of the enzyme activity following the manufacturer’s instructions of a NOS activity assay kit (Merck KGaA, Darmstadt, Germany). After colorimetric reaction, the optical density was read using a microplate spectrophotometer (ThermoFisher Scientific) at an absorbance wavelength of 540 nm. Each sample was analyzed in triplicate and the mean was used for statistical analyses.

### 2.13. Measurement of NADPH Oxidase Activity in RVLM

NADPH oxidase activity of protein samples from RVLM was measured using the lucigenin-derived chemiluminescence method [32]. Briefly, the luminescence assay was performed in phosphate buffer with NADPH as the substrate. After dark adaptation, a tissue homogenate (100 µg protein) was added, and the chemiluminescence value was recorded. O_2_^•^^−^ production was measured with the addition of NADPH, in the presence or absence of an NADPH oxidase, diphenyleneiodonium. All measurements were conducted in the dark room with temperature maintained at 22–24 °C. Light emission was recorded by a Sirius Luminometer (Berthold, Germany). Protein concentrations were determined using a Bio-Rad protein assay kit (Bio-Rad Laboratories). Each sample was analyzed in triplicate and the mean was used for statistical analyses.

### 2.14. Measurement of Total Antioxidant Activity in RVLM

Tissue antioxidant activity in RVLM was measured by a total antioxidant capacity assay kit (Sigma-Aldrich), following the protocol provided by the manufacturer. RVLM tissues were homogenized in lysis buffer, centrifuged, and the supernatant was used for analysis. The reaction was based on Cu^2+^ reduction by the small molecule antioxidants and the reduced Cu^+^ ion chelates with a colorimetric probe that was read with a standard 96-well spectrophotometric microplate reader at 570 nm. Antioxidant capacity was determined by comparison with Trolox, a water-soluble vitamin E analog that serves as an antioxidant standard. Each sample was analyzed in triplicate and the mean was used for statistical analyses.

### 2.15. Measurement of Mitochondrial Respiratory Enzyme Activity in RVLM

The mitochondrial fraction from RVLM tissue was isolated following procedures reported previously [42]. Purity of the mitochondrial-rich fraction was verified by the expression of the mitochondrial cytochrome *c* oxidase. Activities of mitochondrial respiratory chain enzymes were measured immediately after mitochondrial isolation, according to procedures reported previously [42] using a thermostatically regulated spectrophotometer (ThermoFisher Scientific). Enzyme activity was expressed in nmol/mg protein/min.

For the measurement of nicotinamide adenine dinucleotide (NADH) cytochrome *c* reductase (NCCR; enzyme for electron transport between ETC Complex I and Complex III) activity, mitochondrial fraction was incubated in a mixture containing K_2_HPO_4_ buffer, KCN, β-NADH, and rotenone at 37 °C for 2 min. After the addition of cytochrome *c* (50 µM), the reduction of oxidized cytochrome *c* was measured as the difference in the presence or absence of rotenone at 550 nm for 3 min at 37 °C.

For the determination of succinate cytochrome *c* reductase (SCCR; enzyme for electron transport between ETC Complex II and Complex III) activity, mitochondrial fraction was performed in the same buffer solution supplemented with succinate. After a 5 min equilibration at 37 °C, cytochrome *c* (50 µM) was added and the reaction was monitored at 550 nm for 3 min at 37 °C.

For the determination of cytochrome *c* oxidase (CCO, marker enzyme for ETC Complex IV) activity, mitochondria fraction was pre-incubated at 30 °C for 5 min in K_2_HPO_4_ buffer, then 45 µM ferrocytochrome *c* was added to start the reaction, which was monitored at 550 nm for 3 min at 30 °C. In all measurements, experiments were performed in triplicate, and the mean was used for statistical analyses.

### 2.16. Measurement of ATP Levels in RVLM

RVLM tissues were centrifuged at 10,000× *g* for 10 min after homogenization in a protein extraction solution (Pierce, Rockford, IL, USA), ATP concentration in the supernatant was determined by an ATP colorimetric assay kit (AbCam, Waltham, MA, USA) using a microplate reader (ThermoFisher Scientific). The ATP level was normalized to the protein concentration of the sample. Each measurement was performed in triplicate and the mean was used for statistical analyses.

### 2.17. Western Blot Analysis

We determined the expression levels of gp91^phox^, p22^phox^, p67^phox^, and p47^phox^ subunits of NADPH oxidase, manganese dismutase (SOD2), nuclear factor erythroid 2-related factor 2 (Nrf2), nNOS, iNOS, eNOS, NADPH oxidase activator 1 (Noxa1), uncoupling protein 3 (UCP3), and GAPDH in total protein extracted from RVLM by Western blot analysis [43]. In brief, 8–12% SDS-polyacrylamide gel electrophoresis was used for protein separation, and a Bio-Rad miniprotein-III wet transfer unit (Bio-Rad) was employed to transfer samples onto polyvinylidene difluoride transfer membranes (Immobilon-P membrane; Millipore, Bedford, MA) for 1.5 h at 4 °C. The transfer membranes were then incubated with a blocking solution (5% nonfat dried milk dissolved in Tris-buffered saline-Tween buffer) for 1 h at room temperature. Primary antisera used in this study included goat polyclonal, rabbit polyclonal or monoclonal, or mouse monoclonal antiserum against gp91^phox^ (1:5000; BD Biosciences, Sparks, MD, USA), p22^phox^, p67^phox^, and p47^phox^ (1:5000; Santa Cruz Biotechnology, Santa Cruz, CA, USA), SOD2 (1:3000; Stressgen, San Deigo, CA, USA), Nrf2 (1:1000; Santa Cruz), nNOS, iNOS, and eNOS (1:1000; BD Biosciences), Noxa1 (1:1000, Santa Cruze), UCP3 (1:1000, Santa Cruz), and GAPDH (1:10,000; Merck). Membranes were subsequently washed three times with TBS-t buffer, followed sequentially by incubation with the secondary antibodies (1:10,000; Jackson ImmunoResearch, West Grove, PA, USA) for 1 h and horseradish peroxidase-conjugated goat anti-rabbit IgG or goat anti-mouse IgG (Jackson ImmunoResearch). An enhanced chemiluminescence Western blot detection system (GE Healthcare Bio-Sciences Corp., Piscataway, NJ, USA) was used to detect specific antibody–antigen complex. For the detection of eNOS or nNOS dimerization, nondenaturing, low-temperature sodium dodecyl sulfate polyacrylamide gel electrophoresis was used [32,34]. During the electrophoresis process and transfer of proteins to nitrocellulose membrane, buffers were placed in an ice-water bath and the whole apparatus was kept at 4 °C. ImageJ software (NIH, Bethesda, MD, USA) was used to quantify the number of detected proteins, which was expressed as the ratio to loading control (GAPDH).

### 2.18. Generation of Lentiviral Vector

NOXa1 shRNA lentiviral particles (sc-150038-V, Santa Cruz) were used in gene silencing experiments. These transduction-ready viral particles contain a target-specific construct that encodes a 19–25 nt (plus hairpin) shRNA designed to knock down gene expression of *Noxa1*. Each vial contains 200 µL frozen stock of 1.0 × 10^6^ infectious units of virus (IFU) in Dulbecco’s Modified Eagle’s Medium with HEPES pH 7.3 (25 mM). Control shRNA lentiviral particles (sc-108080, Santa Cruz) contain an shRNA construct that encodes a scrambled sequence that will not lead to the specific degradation of any known cellular mRNA.

### 2.19. Microinjection of Lentiviral Vectors into RVLM

Microinjection of the Lv-Noxa1-shRNA, or scramble (Lv-scr-RNA), was carried out stereotaxically and sequentially into the bilateral RVLM of rats that were anesthetized with sodium pentobarbital (50 mg/kg, i.p.). Adequate anesthesia of animals was confirmed by observations of unresponsive to paw pinch and no corneal withdrawal reflex. The animals were placed into a stereotaxic head holder (Kopf, Tujunga, CA, USA) on a thermostatically controlled heating pad. Bilateral microinjection of the viral vectors was carried out, as described previously [32,44]. In brief, a glass micropipette (external tip diameter: 50–80 µm), connected to a 0.5-µL Hamilton microsyringe, was positioned into RVLM. A total of eight injections (4 on each side) of undiluted viral particles (200 nl total volume on each side) were made at two rostro-caudal levels at stereotaxic coordinates of 4.5–5.0 mm posterior to lambda, 1.8–2.1 mm lateral to the midline, and 8.0–8.5 mm below the dorsal surface of cerebellum. These coordinates cover the confines of RVLM within which sympathetic premotor neurons reside [25,40]. After the lentivirus injection, the wound was closed in layers, and animals were allowed to recover in individual cages with free access to food and water.

### 2.20. Reverse Transcription and quantitative Polymerase Chain Reaction

Total RNA from RVLM tissues was isolated with TRIzol reagent (Invitrogen, Carlsbad, CA, USA) according to the manufacturer’s protocol. All RNA isolated was quantified by spectrophotometry and the optical density 260/280 nm ratio was determined. Reverse transcriptase (RT) reaction was performed using a SuperScript Preamplification System (Invitrogen) for the first-strand cDNA synthesis.

*Nox**a1* and *Ucp3* mRNA levels were analyzed by quantitative polymerase chain reaction (qPCR) using SYBR Green and normalized to the GAPDH mRNA signal as described [44]. The following primers were used: *No**xa1*: 5′-TTA CTC TGC CCC TGA AGG TC-3′ (forward) and 5′-CTC GGG CTT TGT TGA AC-3′ (reverse); *Ucp3*: 5′-TTC CTG GGG GCC GGC ACT G-3′ (forward) and 5′-CAT GGT GGA TCC GAG CTC GGT AC-3′ (reverse) [45]; and *GAPDH*: 5′-AGA CAG CCG CAT CTT CTT GT-3′ (forward), 5′-CTT GCC GTG GGT AGA GTC AT-3′ (reverse). Noxa1 and *Ucp**3* mRNA were amplified under the following conditions: 95 °C for 3 min, followed by 50 cycles consisting of 95 °C for 10 s, 50 °C for 20 s, and 72 ° C for 2 s, and finally a 10 min extension at 40 °C. GAPDH was amplified under identical conditions, with the exception of a 55 °C primer annealing temperature. All samples were analyzed in triplicate. All qPCR reactions were followed by dissociation curve analysis. Relative quantification of gene expression was performed using the 2^ΔΔCT^ method.

For amplification of oxidative stress-related mRNA, RT² Profiler PCR Arrays (Qiagen GmbH, Hilden, Germany) were employed following the manufacturer’s protocol. The microarrays include primer assays for 84 oxidative stress-focused genes, 5 housekeeping genes, a genomic DNA control, 3 wells containing reverse-transcription controls, and 3 wells containing a positive PCR control (Appendix A). Total RNA from RVLM tissues was converted into first-strand cDNA using the RT^2^ First Strand Kit. The cDNA was next mixed with an appropriate RT^2^ SYBR^®^ Green Mastermix. This mixture was then aliquoted into the wells of the RT^2^ Profiler PCR Array. PCR was performed, and the relative expression was determined using data from the real-time cycler and the 2^∆∆CT^ method.

### 2.21. Experimental Design

Figure 1 illustrates the experimental design of the present study. The first group of young and adult rats (*n* = 6 per group) was used to evaluate the effect of i.p. l-NAME on SBP, HR, LF component of SAP signals, plasma NE and serum NO levels, plasma MAD and proinflammatory cytokine, and tissue ROS levels and expression of proteins for the production and degradation of ROS in RVLM. The hemodynamic parameters were recorded on days 3 and 1 prior to, and on the day (day 0) before, osmotic pump implantation and days 1, 3, 5, 7, 9, 11, and 14 following i.p. infusion of l-NAME or 0.9% saline. NE and NO were measured on day 3 before, and days 0, 7, and 14 following l-NAME treatment. At the end of the 14-day infusion, the power density of the LF component of SAP signals was determined before animals were killed to collect blood for proinflammatory cytokine measurements, and collect RVLM tissue for measurements of ROS and protein expressions.

The protocol was repeated in a second group of young and adult animals to evaluate various treatments (*n* = 5 per group; see below) on hemodynamic and/or biochemical changes induced by i.p. l-NAME infusion. The pharmacological manipulations, including oral gavage or i.c. infusion, were performed from days 7 to 14 during the 14-day l-NAME treatment period. BP was recorded on day 3 before, and on days 0, 3, 7, 9, 11, and 14 following l-NAME infusion, and power density of LF component, tissue levels of ROS and NO in RVLM were determined at the end of the treatment period.

The third group of young and adult animals was used to identify candidate genes discriminately expressed in RVLM of adult animals (*n* = 3 per group), and the functional significance of the identified mRNA in susceptibility to hypertension induced by l-NAME (*n* = 6 per group) in adult animals. Gene manipulation was carried out via bilateral microinjection of lentiviral vectors into RVLM on day 10 following l-NAME infusion, and mRNA and protein expressions, as well as SBP were measured at the end of the 14-day infusion.

Treatments employed in the present study included i.p. infusion of l-NAME (10 mg/kg/day; Sigma-Aldrich, MA, USA), oral intake via gavage of L-arginine (2%, Sigma-Aldrich), 4-hydroxy-2,2,6,6-tetramethylpiperidine-1-oxyl (tempol; 100 µmol/kg; Sigma-Aldrich), or amlodipine (10 mg/kg; Sigma-Aldrich); or i.c. infusion of L-arginine (2 µg/kg/day), tempol (1 μmol//h/μL), or mitoQ_10_ (2.5 μmol//h/μL; Sigma-Aldrich); or microinjection bilaterally into RVLM of Lv-Noxa1-shRNA (1 × 10^5^ IFU per animal) or Lv-scr-shRNA (1 × 10^5^ IFU per animal). Control infusion of 0.9% saline (for i.p. or oral gavage treatment) or artificial CSF (aCSF; for i.c. infusion) served as the volume and vehicle control. The composition of aCSF was (mM): NaCl 117, NaHCO_3_ 25, Glucose 11, KCl 4.7, CaCl_2_ 2.5, MgCl_2_ 1.2, and NaH_2_PO_4_.

### 2.22. Statistical Analysis

All data were presented as mean ± standard deviation (SD). The normality of the data distribution was checked before all the statistical analyses using Shapiro–Wilk test to confirm that the data complied with normal distribution. Differences in SBP and HR to various treatments were analyzed with a two-way analysis of variance (ANOVA) with repeated measures, followed by the Tukey multiple comparisons test using time and treatment group as the main factors. All the other differences in mean values were analyzed by one-way ANOVA with Tukey’s multiple comparisons tests. Statistical differences between experimental groups in young and adult animals were evaluated using unpaired Student’s *t*-tests. All the data were analyzed by GraphPad Prism software (version 6.0; GraphPad Software Inc., La Jolla, CA, USA). *p* < 0.05 was considered statistically significant.

## 3. Results

### 3.1. Age-Dependent Changes in Blood Pressure, Heart Rate, Sympathetic Vasomotor Activity, and Plasma NE Levels in Response to Systemic NO Deficiency

Our first set of experiments evaluated the age-dependent hemodynamic responses to systemic NO deficiency, a well-established animal model for the study of human hypertension [16]. In young (at age of 8 weeks) normotensive WKY rats, i.p. infusion of l-NAME (10 mg/kg/day) for 14 days resulted in gradual increases in SBP, power density of the LF component of the SAP signals, our experimental index for neurogenic sympathetic vasomotor activity [32,43], and plasma NE levels, but not HR, which became statistically significant on postinfusion days 7−14 (Figure 2A−D). Similar cardiovascular responses were observed in adult animals (at age of 20 weeks). Of note, baseline SBP (105 ± 4.4 versus 90.7 ± 5.4 mmHg, *n* = 6, *p* < 0.05), as well as temporal increases in SBP (139.8 ± 3.5 versus 125.5 ± 6.5 mmHg, *n* = 6; *p* < 0.05), LF power density (2.76 ± 0.26 versus 2.39 ± 0.24 mmHg^2^, *n* = 6, *p* < 0.05), and plasma NE levels (7.16 ± 0.33 versus 6.40 ± 0.46 ng/mL, *n* = 6, *p* < 0.05) measured on day 14 postinfusion, were significantly greater in adult animals when compared to young rats. l-NAME infusion evoked similar decreases in serum NO (nitrite and nitrate) levels, detected on days 7 and 14 postinfusion in both age groups (Figure 2E). The same treatment, on the other hand, exerted no effect on plasma Il-1β, IL-6, and TNF-α levels measured at the end of l-NAME infusion (Table 1). These data suggest an age-dependent vulnerability in hemodynamic changes associated with systemic NO deficiency.

### 3.2. Effect of Systemic l-NAME Treatment on Expression of NOS Isoforms and Activity in RVLM

l-NAME reportedly crosses the BBB to alter NOS expression and inhibit NOS activity in brain [22,23,24]. At the end of 14-day i.p. infusion of l-NAME, protein expression of eNOS, but not nNOS or iNOS, isoform in RVLM was significantly decreased (Figure 3A), alongside a notable suppression of NOS activity (Figure 3B) in both young and adult animals. Of note, systemic l-NAME treatment resulted in comparable suppression in eNOS expression and total NOS activity in RVLM of both age groups.

### 3.3. Differential Effect of Systemic l-NAME Treatment on Tissue ROS Levels in RVLM of Animals at Different Ages

A series of studies from our laboratory [32,39,42,43] suggest that the LF component of SAP signals originates from the RVLM, and tissue oxidative stress in RVLM augments sympathetic vasomotor activity and BP [28,32,39,41,42,43,44]. We therefore investigated whether the differential effect of l-NAME infusion on hemodynamic parameters at different ages is the consequence of disparate tissue oxidative stress in RVLM. As shown in Figure 4A, baseline tissue ROS levels were higher, albeit statistically insignificant, in RVLM of adult animals. Moreover, i.p. l-NAME (10 mg/kg/day) infusion for 14 days resulted in further increases in tissue ROS levels in RVLM of adult, but not young rats. On the other hand, there were no detectable increases in plasma MDA levels (a biomarker of oxidative stress) in both groups (Figure 4B).

Dysregulated redox homeostasis because of an imbalance in ROS production over degradation leads to tissue oxidative stress [6,26,28,33]. In RVLM, we reported previously that increases in the protein expression of NADPH oxidase subunits [43], and decreases in the expression of antioxidants [46], contribute to oxidative stress that results in sympathoexcitation and hypertension in spontaneously hypertensive rats and normotensive animals treated with angiotensin II (Ang II). We therefore examined the expression of NADPH oxidase subunits and antioxidants in RVLM of animals that were subjected to i.p. l-NAME infusion. In RVLM of WKY rats at age of 8 weeks, the protein expression of pg91^phox^ and p22^phox^, but not p47^phox^ or p67^phox^ subunit of NADPH oxidase (Figure 4C), determined on day 14 following i.p. infusion of l-NAME, was appreciably increased. Interestingly, the protein expression of two key antioxidants, SOD2 and Nrf2 (Figure 4D), was also increased at the same postinfusion time point. Similar results were found in RVLM of l-NAME-treated animals at the age of 20 weeks. As shown in Figure 4E, systemic l-NAME treatment also significantly augmented the enzyme activity of NADPH oxidase and total antioxidant capacity in RVLM, measured on day 14 postinfusion in both age groups. Notably, the increase in NADPH oxidase activity in RVLM was significantly greater in the adult animals (+66.5 ± 13.8% versus +30.5 ± 10.2 %, *n* = 6, *p* < 0.05).

Mitochondria are considered another important cellular source of ROS. In RVLM, impairment of enzyme activity of the mitochondrial electron transport chain (ETC) for oxidative phosphorylation contributes to cellular oxidative stress, leading to sympathetic hyperactivity and neurogenic hypertension [42]. In young normotensive rats, systemic l-NAME treatment increased the enzyme activity of CCO (electron transport in Complex IV), but not NCCR (enzyme for electron transport between Complexes I and III) or SCCR (enzyme for electron transport between Complexes II and III) in RVLM (Figure 5A), accompanied by a mild increase in tissue ATP content (15.8 ± 2.1 versus 17.8 ± 1.3 pmol/µg, *n* = 6, *p* = 0.071). On the other hand, the enzyme activity of both NCCR and CCO was notably depressed in RVLM of control adult rats, and remained reduced following l-NAME treatment; together with a moderate decrease in tissue ATP content (15.4 ± 1.3 versus 13.2 ± 1.3 pmol/µg, *n* = 6, *p* < 0.05).

The increase in ROS production in RVLM could also result from NOS uncoupling [32]. In RVLM of the l-NAME-treated animals, the ratio of eNOS monomer over dimer, an experimental index of NOS uncoupling, remained unchanged in both age groups (Figure 5B); although, eNOS expression, NOS activity, and tissue NOx levels were suppressed. Similarly, the same treatment did not affect nNOS coupling, which has been reported to promote tissue oxidative stress in RVLM [32], in both age groups.

We interpret our observations that tissue ROS levels, particularly after i.p. l-NAME infusion, were significantly augmented in RVLM of adult rats to suggest an active role of ROS in RVLM in age-dependent exacerbation of hemodynamic responses in this NO deficiency model of hypertension. The elevated protein expression and enzyme capacity of the antioxidants may explain why redox homeostasis in RVLM of young rats is maintained despite the increase in both protein expression and enzyme activity of NADPH oxidase after l-NAME treatment. On the other hand, the enhanced augmentation of NADPH oxidase activity may account for the further increase in tissue ROS levels observed in RVLM of adult animals. The reduced mitochondrial bioenergetics because of impaired enzyme activity of NCCR and CCO may also give rise to the heightened ROS levels in RVLM of adult animals after systemic l-NAME treatment.

### 3.4. Causal Involvement of Tissue Oxidative Stress in RVLM in Age-Dependent Exacerbation of Hemodynamic Responses to Systemic NO Deficiency

To ascertain a causal disparate role of NO deficiency and ROS production in RVLM in age-dependent augmentation of hemodynamic responses to i.p. l-NAME infusion, young and adult animals were randomly divided into six groups to receive, respectively, an oral intake or i.c. infusion of L-arginine, a NOS precursor; tempol, a ROS scavenger; mitoQ_10_, a mitochondrial-targeted SOD mimetic, or amlodipine, a vasodilator of dihydropyridine class of calcium channel blockers; from days 7 to 14 following i.p. infusion of l-NAME (10 mg/kg/day). In young rats, oral intake of L-arginine (2%) or amlodipine (10 mg/kg), or i.c. infusion of L-arginine (2 µg/kg/day), but not oral intake of tempol (100 µmol/kg) or i.c. infusion of tempol (1 μmol//h/μL) or mitoQ_10_ (2.5 μmol//h/μL), significantly attenuated the increase in SBP in the l-NAME-treated animals (Figure 6A). On the other hand, when L-arginine, tempol, or mitoQ was microinfused into the cisterna magna of adult animals, one week after i.p. l-NAME infusion, the treatments discernibly diminished the increases in SBP induced by systemic l-NAME treatment (Figure 6B). Microinfusion of tempol or mitoQ_10_, but not L-arginine, into the cisterna magna, at the same time, restored tissue ROS in RVLM of the l-NAME-treated rats to saline-control levels (Figure 6C). Infusion of tempol or mitoQ_10_ into the cisternal magna had no apparent effect on tissue NO levels in RVLM (Figure 6D), and i.c. infusion of tempol or mitoQ_10_ infusion did not affect the reduced serum NO levels following l-NAME infusion.

These results are interpreted to suggest that in response to systemic NO deficiency, a predominant increase in vasomotor tone because of vascular constriction and sympathetic outflow from RVLM may underline the increase in SBP of the l-NAME-treated young animals. When animals become older, tissue oxidative stress in RVLM is actuated to further increase sympathetic vasomotor activity and promote greater hemodynamic responses in the systemic l-NAME treatment model of hypertension.

### 3.5. Identification of Additional Age-Dependent Redox Homeostasis-Related Genes in RVLM in the Systemic NO-Deficiency Model of Hypertension

Our observations that the increase in tissue ROS levels was greater in RVLM of the l-NAME-treated adult animals (cf. Figure 4E) prompted the speculation that additional oxidative stress-related genes on top of those reported previously are upregulated in this brain stem site. To address this issue, we performed a whole genome microarray analysis of RVLM tissue using a Qiagen RT^2^ Profiler^TM^ PCR array (Qiagen). As shown in the supplementary data (Appendix A), four differentially expressed genes, whose expression levels are at least two times different from young rats, were identified in RVLM of adult animals. These included two upregulated genes, NADPH oxidase activator 1 (*Noxa1*) and glutathione peroxidase 2 (*GPx2*), and two downregulated genes, dual oxidase 2 (*Duox2*) and uncoupling protein 3 (*Ucp3*). Given the limitation in the RVLM sample volume, we decided to verify the accuracy of our microarray analysis by real-time qPCR only on the candidate transcriptomes that exhibited the highest (Noxa1) and lowerest (*Ucp3*) changes. Our RT-qPCR results confirmed the microarray data for *Noxa1* and *Ucp3* mRNA in RVLM (Figure 7A). Moreover, their expressions were further upregulated in the l-NAME-treated animals in both age groups. Similar patterns were found in protein expression of Noxa1 and UCP3 in RVLM of both groups of animals (Figure 7B). Of note, the upregulation of *Noxa1* (+1.9 ± 0.2 versus +1.1 ± 0.5 fold, *n* = 6, *p* < 0.05), but not a change in *Ucp3* mRNA (+1.7 ± 0.4 versus +1.8 ± 0.3 fold, *n* = 6, *p* > 0.05), was significantly greater in RVLM of adult animals when compared to young rats. Together these results suggest that age-dependent alterations in oxidative stress-related gene transcription, including the upregulation of a ROS-producing enzyme, *Nox1a*, and the downregulation of an antioxidant, *Ucp3*, may contribute to the elevated ROS levels in RVLM following i.p. l-NAME infusion.

### 3.6. Silencing Nox1a mRNA in RVLM Ameliorates Oxidative Stress and Attenuates Hemodynamic Responses to Systemic NO Deficiency in Adult Rats

Our final series of experiments was performed to validate the functional significance of the newly identified *Noxa1* gene in RVLM on exacerbated hemodynamic responses in the l-NAME-treated adult animals. In situ gene silencing via bilateral microinjection into RVLM of lentiviral vectors encoding shRNA targeting *Noxa1* (Lv-Noxa1-shRNA; 1 × 10^5^ IFU) was performed on day 10 after the onset of i.p. l-NAME infusion. Effective transduction of the viral vectors into RVLM was confirmed by qPCR, which showed a significant decrease in *Noxa1* mRNA and protein, measured on day 4 after vector transduction (Figure 8A). Compared to their scramble shRNA control, silencing *Nox1a* in RVLM with its shRNA resulted in notable attenuation of hypertension as well as the increase in sympathetic vasomotor activity and plasma NE levels (Figure 8B) in the l-NAME-treated adult animals. Gene silencing of *Noxa1* also significantly alleviated the elevated ROS levels (Figure 8C) in RVLM, but not the reduced plasma NO levels (Figure 8D). These data provide evidence to suggest an active role of *Noxa1* in RVLM on age-dependent susceptibility to hypertension induced by systemic l-NAME treatment in adult rats.

## 4. Discussion

The present study was designed to explore the role of oxidative stress in RVLM on age-dependent susceptibility to hypertension in response to systemic NO deficiency, and to decipher the underlying molecular mechanisms. There are four major findings. First, i.p. infusion of l-NAME evoked oxidative stress in RVLM in adult, but not young, normotensive rats, accompanied by augmented enzyme activity of NADPH oxidase and reduced mitochondrial NCCR and CCO enzyme activities. Second, treatment with L-arginine via oral gavage or infusion into the cistern magna, but not i.c. tempol or mitoQ_10_, significantly offset the l-NAME-induced hypertension in young rats. On the other hand, all treatments appreciably reduced l-NAME-induced hypertension in adult rats. Third, four genes involved in ROS production and clearance were differentially expressed in RVLM in an age-related manner. Of them, *Noxa1* and *GPx2* were upregulated and *Duox2* and *Ucp3* were down-regulated. Systemic l-NAME treatment caused greater upregulation of *Noxa1*, but not *Ucp3*, mRNA expression in RVLM of adult rats. Fourth, gene silencing of *Noxa1* in RVLM effectively alleviated oxidative stress and protected adult rats against l-NAME-induced hypertension. These data together suggest that hypertension induced by systemic l-NAME treatment in young rats is mediated primarily by NO deficiency that occurs both in vascular smooth muscle cells and RVLM. On the other hand, enhanced augmentation of oxidative stress in RVLM contributes to a heightened susceptibility of adult rats to hypertension induced by systemic l-NAME treatment.

NO deficiency is a well-characterized trait in human hypertension [5,6,7], and NOS inhibition by l-NAME is commonly used to establish NO deficiency in animal models of human hypertension [16]. l-NAME exerts various pharmacological effects on cardiovascular functions and molecular activities that are dependent on dose (1–50 mg/kg/day), route (oral, subcutaneous, i.p., or cerebral ventricle), and mode (acute, daily bolus, or continuous infusion) of administration, as well as the duration of treatment (minutes, hours, days, or weeks). In this study, we employed a relatively low dose for i.p. infusion of l-NAME (10 mg/kg/day) for 2 weeks to establish the expected hemodynamic responses, with minimal concomitant oxidative and inflammatory actions (cf. Table 1) to avoid their confounding influences on hypertension development. At a higher dosage and/or longer duration of l-NAME treatment, ROS production in vascular smooth muscle cells [47], kidney [48], and heart [21], as well as pro-inflammatory cytokines in kidney [49] and heart [21], have been demonstrated to mediate hypertension and associated cardiovascular complications such as cardiac hypertrophy and renal injury. The findings that the plasma level of NE was increased and serum NO level was reduced, and that oral intake of L-arginine and amlodipine (cf. Figure 6A,B) conferred protection against l-NAME-induced hypertension suggest that an increase in SNA and vasoconstriction may contribute to the observed hemodynamic changes. We did not find significant changes in HR following l-NAME administration; although, other studies demonstrated reduction [50] or augmentation [18] in HR. The exact reason behind these diverse findings is not clear, but might simply be the consequence of differences in dose and duration of l-NAME treatment. In this regard, l-NAME given at 1.5 times our dose for 2 weeks decreases [50], and at 4 times our dose for 5 weeks increases, HR [18].

To date, cardiovascular responses to l-NAME treatment have primarily focused on its effects on NOS expression and NO bioavailability in the vasculature, with little attention on the role of l-NAME in the central nervous system. As such, one of the major findings of the present study is the identification of the suppressive effects of systemic l-NAME treatment on eNOS protein expression, NOS activity, and tissue NO level in RVLM of both young and adult rats. l-NAME has been reported to cross the BBB to reach brain tissues [22,23,24]. All three NOS isoforms are constitutively present in RVLM [51], and their roles in RVLM on neural control of cardiovascular functions, have been extensively reviewed in the literature [26]. In the present study, we found that the expression of eNOS, but not nNOS or iNOS, protein in RVLM was suppressed by i.p. l-NAME infusion. Since constitutive eNOS tonically inhibits RVLM neuronal activity and sympathetic outflow [52], a diminished eNOS expression and eNOS-derived NO availability in RVLM may therefore contribute to the observed increases in sympathetic vasomotor activity and plasma NE levels in rats subjected to systemic l-NAME treatment. These findings are in concordance with the observations that inhibition of eNOS by l-NAME evokes central sympathoexcitation, leading to increased SNA in experimental animals [19] and healthy men [53]. In addition, diminished eNOS expression and eNOS-derived NO bioavailability in the hypothalamic paraventricular nucleus (PVN) by l-NAME contributes to sympathoexcitation and hypertension associated with heart failure [54]. We reported previously a concentration-dependent action of NO in RVLM on neural control of cardiovascular functions. Whereas nNOS-derived NO is responsible for sympathoexcitation, iNOS-induced NO elicits sympathoinhibition [51]. In contrast, a sympathoinhibitory action of NO derived from nNOS in RVLM has also been reported [29]. In the present study, neither nNOS nor iNOS expression in RVLM was affected by systemic l-NAME treatment. These results are at variance with a previous study [24] that shows nNOS mRNA in RVLM is downregulated in young (4 weeks), but upregulated in adult (10 weeks) rats, following oral intake of l-NAME (50 mg/kg/day) for 6 weeks. Such discrepancies may again reflect differences in dose, route, and duration of l-NAME treatment.

In addition to diminished eNOS/NO signaling, we found in the present study that systemic l-NAME treatment also affected proteins involved in ROS production and clearance in RVLM. Of the major sources for the production of ROS, we found the protein expression of gp91^phox^ and p22^phox^ subunits was increased in both age groups, alongside elevated enzyme activity of the NADPH oxidase. In addition, the enzyme activity of mitochondrial CCO was increased in young, but decreased in adult, rats, and NCCR activity was also decreased in adult rats. On the contrary, NOS uncoupling, which is considered a secondary source of ROS production [6], was not affected by systemic l-NAME treatment. The NADPH oxidase family is the most important enzymatic source of ROS in the cardiovascular system [6,26]. In RVLM, augmented ROS production resulting from increases in pg91^phox^ and P22^phox^ subunits initiates a series of molecular events, leading to tissue oxidative stress and sympathoexcitation that contribute to the neural mechanism of hypertension [43]. Expression of p47^phox^ and p67^phox^ proteins, the other two NADPH oxidase subunits that play active roles in the redox-associated neural mechanism of hypertension [43], on the other hand, were not affected by l-NAME treatment; this might be related to the susceptibility of individual subunits to NO deficiency. The mechanisms underpinning the increase in gp91^phox^ and p22^phox^ protein expressions induced by l-NAME are not immediately clear, but might be the consequence of transcriptional upregulation of these subunits [14,24].

Another major cellular source of ROS production is the mitochondrial ETC in association with oxidative phosphorylation for ATP synthesis [55]. The effects of NO on mitochondrial functions and metabolism are mediated mainly through their interactions at specific sites in the ETC enzyme complexes. In this regard, NO, at subnanomolar amounts, inhibits Complex IV via interactions with the ferrous heme iron or oxidized copper at the heme iron:copper binuclear center of the enzyme [56]. At high concentrations, NO inhibits Complex I via oxidation or *S*-nitrosation of specific thiols [57]. Accordingly, an increase in CCO activity observed in RVLM of l-NAME-treated young rats could result from the withdrawal of an inhibitory effect of NO on Complex IV. On the other hand, the mechanism that underlies the lower NCCR and CCO activity in RVLM of adult rats and their reduced responsiveness to NO deficiency is unclear. Nonetheless, it is noteworthy that aging selectively downregulates genes encoding Complex I and III of the mitochondria ETC both in rat and human hearts [58]. Functionally, impairment of both NCCR and CCO activity in RVLM has been reported to increase mitochondrial ROS production that contributes to hypertension in SHR or Ang II treatment in normotensive rats [46].

Under the condition of oxidative stress, NOS may remove an electron from NADPH and donate it to an oxygen molecule for generation of O_2_^•^^−^ rather than NO [6,14,59]. In RVLM, tissue oxidative stress causes an uncoupling of eNOS during hypertension [28], further depleting the levels of NO and aggravating hypertension progression. In addition, a redox-sensitive feedforward mechanism of nNOS uncoupling in RVLM contributes to sympathoexcitation and hypertension associated with metabolic disorders [32]. In the present study, the ratio between dimmers over monomers of either eNOS or nNOS was not affected (cf. Figure 5B) by i.p. l-NAME infusion, suggesting a negligible role of NOS uncoupling in ROS production in RVLM in response to systemic NO deficiency.

Redox homeostasis depends on the balance between the production and degradation of the oxidants. At the same time, antioxidant treatments offset the development of l-NAME-induced hypertension by a reduction in ROS production during NOS inhibition [50]. In young and adult rats subjected to systemic l-NAME treatment, we found that the protein expression of two key antioxidants, SOD2 and Nrf2, was significantly increased, alongside an increase in total antioxidant activity in RVLM. SOD2, or manganese SOD, is one of the most well-characterized antioxidant defensive mechanisms for the elimination of cellular oxidants, particularly O_2_^•^^−^. Transcribed from *sod**2* and synthesized in the cytoplasm, SOD2 is subsequently relocated to the mitochondrial matrix, endowed with the responsibility to scavenge O_2_^•^^−^ produced by the mitochondrial ETC [60]. In RVLM, transcriptional upregulation of *sod2* protects against mitochondrial oxidative stress and hypertension in Ang II treatment in normotensive rats [61]. SOD2 also participates in the protection again hypertension and cardiovascular complications conferred by the mitochondria-target antioxidants [62]. Nrf2 is the master regulator of antioxidant genes and, hence, of antioxidant status. Nrf2 has been demonstrated to be a key to redox homeostasis in RVLM; targeted ablation of Nrf2 in RVLM leads to hypertension [41]. The increase in the expression of these antioxidants may explain why redox homeostasis in RVLM of young rats was maintained despite significant elevations in ROS production by mitochondria and NADPH oxidase induced by systemic NO deficiency. These data, at the same time, suggest that the oxidant responsiveness to systemic l-NAME treatment is well tolerated in RVLM of young rats but may turn remitted when animals become older.

Aging is associated with an increase in ROS production, which together with a decline in antioxidant defense efficiency significantly contributes to the manifestation of an oxidative stress state [38]. Compared to young rats, we found in this study greater increases in NADPH oxidase activity and augmented ROS accumulation in RVLM of adult rats in response to systemic NO deficiency. These intriguing findings prompted us to search for additional candidate molecules that are associated with age-dependent oxidative stress in RVLM. Based on microarray analysis of redox signal-related genes, we identified four genes whose expression levels are at least two times up- or downregulated in RVLM of adult animals. We found that *Noxa1* and *Gpx2* mRNA were upregulated, whereas *Duox2* and *Ucp3* mRNA were downregulated. Among them, upregulation of the antioxidant Gpx2 could be an antioxidant defense mechanism to compensate for tissue oxidative stress, and Duox2 is a p22^phox^-independent isoform that is not important in cardiovascular pathophysiology [6]. We therefore focused on Noxa1 and *Ucp3* mRNA to further interrogate their roles in the augmented ROS levels in RVLM of adult rats. First, we confirmed that expression of *Noxa1* mRNA was higher, whereas *Ucp3* mRNA was lower, in RVLM of adult rats. An age-dependent decrease in Ucp3 expression in male mice has recently been reported [63]. Second, expression of *Noxa1*, but not *Ucp3*, mRNA was upregulated by systemic l-NAME treatment, suggesting that transcriptional regulation of ROS signal-related genes as a consequence of tissue NO deficiency is target specific. Third, gene silencing of *Nox1* appreciably alleviated the augmented ROS levels in RVLM, indicating the age-dependent accumulation of ROS in RVLM may be attributed to an upregulation of *Noxa1* transcription. Finally, the functional significance of the newly identified *Noxa1* mRNA in RVLM on age-dependent susceptibility of cardiovascular responses to tissue NO deficiency is validated by our findings that bilateral microinjection into RVLM of Lv-Noxa1-shRNA appreciably ameliorated hypertension, the exaggerated sympathetic vasomotor activity, and plasma NE levels evoked by l-NAME treatment in adult animals. Noxa1 is a critical functional homolog of p67phox for NADPH oxidase activation in vascular smooth muscle cells [64]. NOX1 (a p22^phox^-dependent oxidase) interacts with p67phox homolog Noxa1, causing constitutive production of O_2_^•^^−^ [65]. Conversely, genetic deletion of *Noxa1* reduces basal and Ang II-induced hypertension and renal oxidative stress [66]. The observations of comparable changes in Ucp3 expression in RVLM of young and adult animals to systemic l-NAME treatment (cf. Figure 7) are interpreted to suggest a minor role of mitochondrial Ucp3 in RVLM on age-related susceptibility to hypertension in adult rats to systemic NO deficiency. This suggestion, nonetheless, waits for further validation.

Treatments targeting NO and ROS signals in the periphery and RVLM were employed to further verify the differential roles of NO and ROS in RVLM on age-dependent cardiovascular responses induced by l-NAME. In young rats, both oral intake and i.c. infusion of L-arginine, but not i.c. application of tempol or mitoQ_10_, significantly reduced hypertension induced by systemic NO deficiency. These results indicate that l-NAME-induced cardiovascular responses in young animals may mainly be the result of the NO deficiency that occurs both in the smooth muscle cells and RVLM. The engagement of tissue oxidative stress in RVLM on cardiovascular responses to l-NAME in adult animals was unveiled by observations that apart from L-arginine, i.c. infusion of tempol and mitoQ_10_ significantly diminished hypertension. It is noteworthy that i.c. infusion of tempol or mitoQ_10_ had no effect on the reduced NO levels in RVLM, indicating the protective actions of tempol and mitoQ_10_ are not secondary to changes in tissue NO contents. Moreover, the results that i.c. infusion of L-arginine had no effect on the augmented ROS levels in RVLM of adult rats (cf. Figure 6C) suggesting that aging-associated oxidative stress may be related to changes in ROS signals but not NOS activity or tissue NO bioavailability in RVLM. This notion of a minor role of NO in ROS production in RVLM of adult animals is further supported by findings that l-NAME had no effect on the reduced enzyme activity of mitochondrial NCCR and CCO in RVLM of adult rats (cf. Figure 5A). Finally, the observations that oral intake of L-arginine and amlodipine protected both young and adult rats from l-NAME-induced hypertension suggest that the observed cardiovascular changes are likely the final outcomes of vasoconstriction in response to systemic NO deficiency.

There are several limitations to our study. First, the present findings were made from animals that were subjected to a low-dose l-NAME treatment. As discussed above, in view of the disparity of cardiovascular responses that are dependent on doses of l-NAME, the notion of an interplay between NOS and ROS in the pathogenesis of hypertension induced by systemic NO deficiency should be taken with caution. Second, since the present study focused only on RVLM, the roles of NOS and ROS in other “pre-autonomic” neurons, such as the nucleus tractus solitarii (NTS) and PVN, in neural mechanisms of the l-NAME-induced cardiovascular complications remain to be delineated. In this regard, both NOS and ROS signaling in the NTS and PVN have been reported to play pivotal roles in hypertension induced by systemic l-NAME treatment [6,20,26,30,31,67]. Third, an increase in Ang II release along with depressed NO production is considered the principal culprit in hemodynamic and structural alterations in l-NAME-treated rats [68]. In RVLM, Ang II induces ROS production via activation of NADPH oxidase [6,26,32]. In addition, NO deficiency differentially affects the expression of Ang II receptors in RVLM of young versus adult rats [24]. Therefore, it would be of interest to further investigate the role of Ang II in RVLM in the interplay between NOS and ROS on age-dependent susceptibility to hypertension induced by systemic l-NAME treatment. Fourth, in the present study, we used a commercially available microarray kit to screen the oxidative stress-related genes that are differentially expressed in RVLM of young versus adult rats. The genes provided in the kit are far from complete; consequently, the identified genes could be underestimated. In a recent study, out of 47 genes that are involved in ROS metabolism, 39 are downregulated and 8 upregulated in the aged (24 months) versus adult (6 months) rat heart [58].

## 5. Conclusions

In conclusion, our findings reveal that disparate mechanisms underlie the increase in SNA and BP in rats subjected to systemic l-NAME treatment in an age-dependent manner. In young rats, cardiovascular responses to l-NAME are mediated mainly by NO deficiency, both in the vascular smooth muscle cells and RVLM. When animals become older, additional ROS generation from both mitochondrial (reduction in enzyme activity of NCCR and CCO) and extra-mitochondrial (transcriptional upregulation of Noxa1) pathways may contribute to the enhanced susceptibility to sympathoexcitation and hypertension induced by systemic l-NAME treatment. This information provides novel insights into potential targets involved in the responsiveness to systemic NO deficiency during aging that could be manipulated to prevent age-associated deterioration in cardiovascular functions. Moreover, recognizing the functional significance of aging on the transcription of genes encoding ROS signaling molecules may help to identify novel targets that can be selectively intervened to prevent aging-associated hypertension and cardiovascular complications.

## Figures and Tables

**Figure 1 biomedicines-10-02232-f001:**
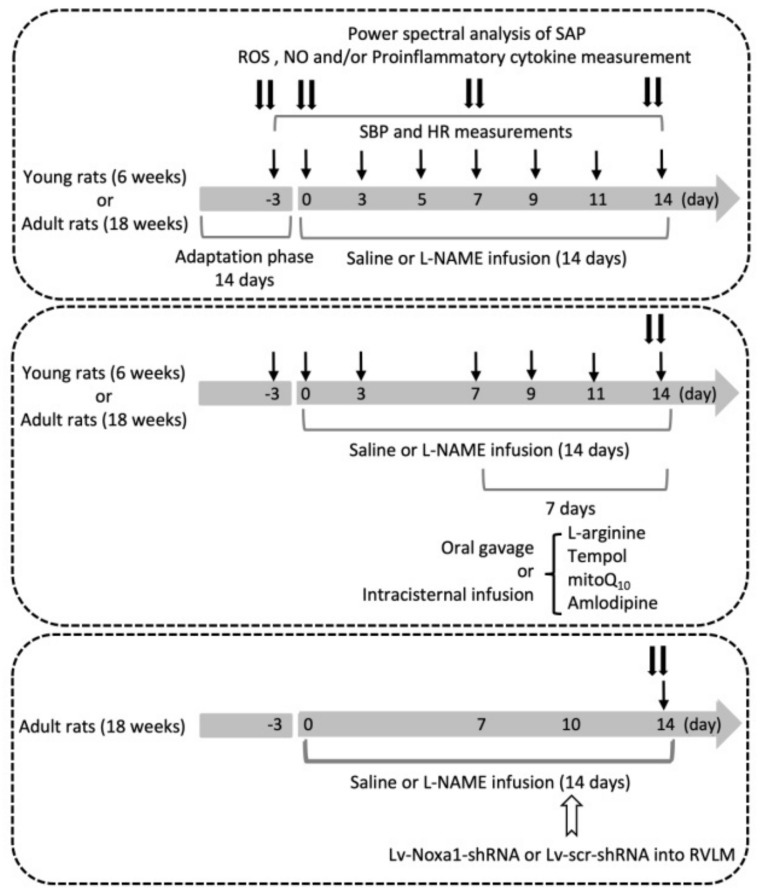
Experimental design of the present study. The first group of young and adult animals (*n* = 6 per group) was used to assess systolic blood pressure (SBP), heart rate (HR), and power density of low-frequency component in the SBP spectrum at various time intervals (arrows) before and after i.p. infusion of l-NAME for 14 days. Nitric oxide (NO), reactive oxygen species (ROS), and/or proinflammatory cytokine in plasma and/or tissue of rostral ventrolateral medulla (RVLM) were measured at the end of the 14-day l-NAME treatment (double arrows). The protocol was repeated in a second group of young and adult rats (*n* = 5 per group) to evaluate the effect of various treatments, delivered via oral gavage or intracisternal infusion during days 7–14, on l-NAME-induced changes in SBP, HR, LF power, and NO and ROS levels in RVLM. The third group of adult animals (*n* = 6 per group) was used to examine the effect of gene silencing NADPH oxidase activator 1 (*Noxa1*) in RVLM on changes in SBP, HR, LF power, and NO and ROS levels in RVLM induced by systemic l-NAME. Lentiviral vector contains a target-specific construct that encodes a short hairpin RNA (shRNA) to knock down gene expression of *Noxa1* (Lv-Noxa1-shRNA) or control scrambled shRNA (Lv-scr-shRNA) was microinjected into the bilateral RVLM (open arrow) on day 10 after l-NAME infusion.

**Figure 2 biomedicines-10-02232-f002:**
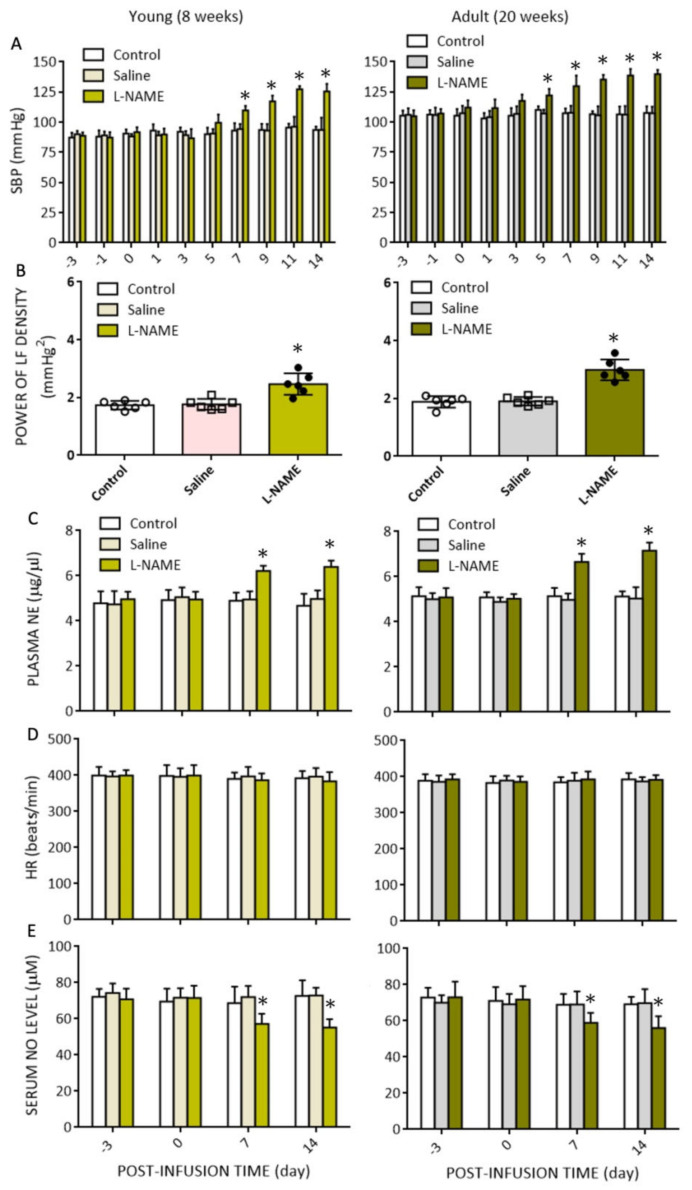
Temporal changes in hemodynamic parameters and serum NO (nitrite and nitrate) levels in response to intraperitoneal infusion of l-NAME (10 mg/kg/day) for 14 days. Changes in (**A**) systolic blood pressure (SBP); (**B**) power density of low-frequency (LF) component of SBP signal; (**C**) plasma norepinephrine (NE) levels; (**D**) heart rate (HR); as well as (**E**) serum NO levels detected at different time points in the untreated group, or animals treated with i.p. infusion of saline or l-NAME. Data are presented as mean ± SD, *n* = 6 per group at each time interval. * *p* < 0.05 versus saline-treated group (pink or gray bars) in post hoc Tukey’s multiple-range test.

**Figure 3 biomedicines-10-02232-f003:**
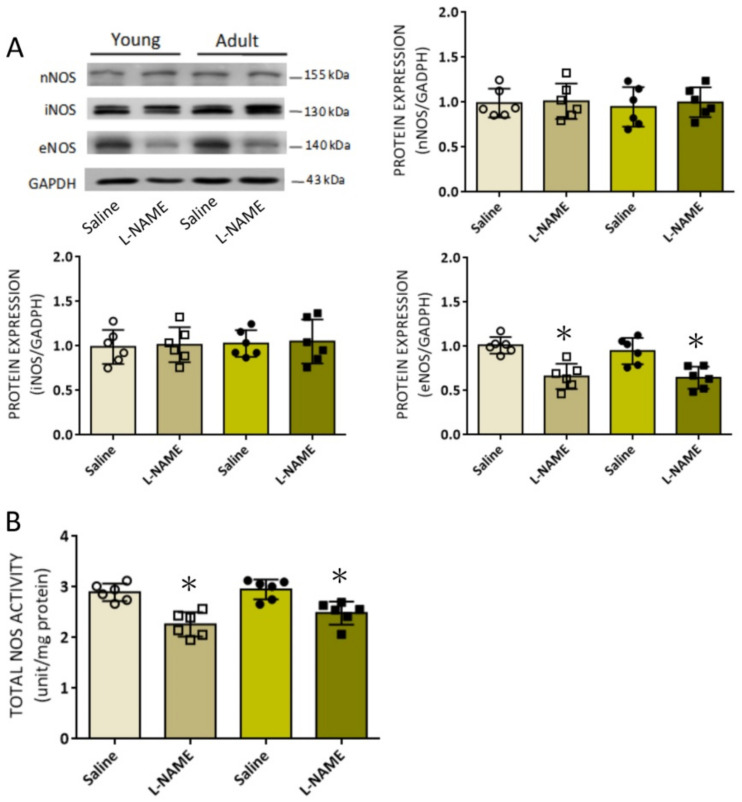
Effect of systemic l-NAME treatment on the expression of NOS isoforms and NOS activity in RVLM: (**A**) Representative gels (insets) and densitometric analysis of results from Western blot analysis showing changes in protein expression of nNOS, iNOS, and eNOS and (**B**) enzyme activity of NOS in RVLM 14 days after i.p. infusion of saline or l-NAME (10 mg/kg/day) in young (8 weeks, open circles or squares) or adult (20 weeks, filled circles or sqaures) rats. Data on protein expression were normalized to the respective saline control value, which is set to 1.0, and are presented as mean ± SD, *n* = 6 per group. * *p* < 0.05 versus corresponding saline-treated group in unpaired Student’s *t*-test.

**Figure 4 biomedicines-10-02232-f004:**
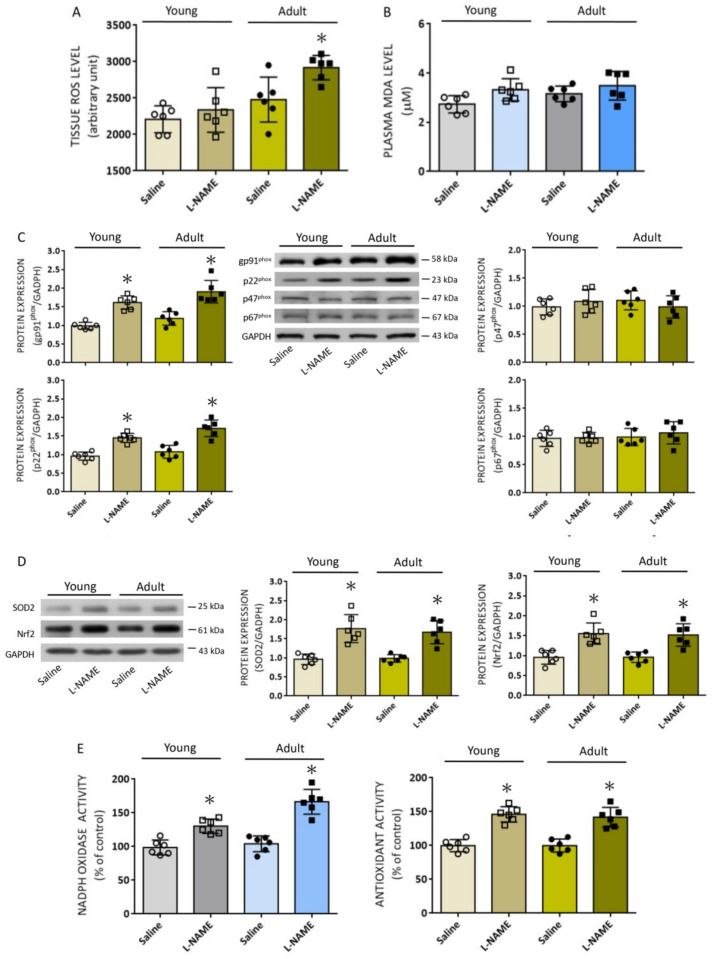
Effect of systemic l-NAME treatment on reactive oxygen species (ROS) levels and expression of proteins involved in ROS production and degradation. Data showing ROS levels in RVLM (**A**) and MDA levels in plasma (**B**) of young (8 weeks, open circles or squares) or adult (20 weeks, filled circles or squares) rats after i.p. infusion of saline or l-NAME (10 mg/kg/day) for 14 days. Also shown are representative gels (insets) and densitometric analysis of results from Western blot changes in protein expression of pg91^phox^, p22^phox^, p47^phox^, and p67^phox^ (**C**) or SOD2 and Nrf2 (**D**), as well as enzyme activity of NADPH oxidase and total antioxidants (**E**) in RVLM 14 days after saline or l-NAME treatment. Data on protein expression are normalized to the respective saline control value, which is set to 1.0. Data are presented as mean ± SD, *n* = 6 per group. * *p* < 0.05 versus corresponding saline-treated group in unpaired Student’s *t*-test.

**Figure 5 biomedicines-10-02232-f005:**
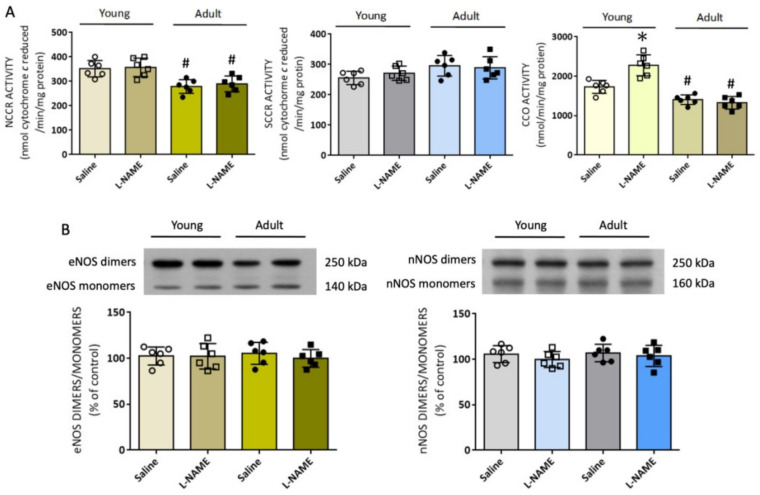
Effect of systemic l-NAME treatment on mitochondrial electron transport chain enzyme activity and NO synthase (NOS) uncoupling in RVLM: (**A**) Enzyme activities of NADH cytochrome *c* reductase (NCCR, marker for coupling capacity between complexes I and III), succinate cytochrome *c* reductase (SCCR, marker for coupling capacity between complexes II and III), and cytochrome *c* oxidase (CCO, marker for complexes IV) in RVLM of young (8 weeks, open circles or squares) or adult (20 weeks, filled circles or squares) rats after i.p. infusion of saline or l-NAME (10 mg/kg/day). (**B**) Representative gels (insets) and densitometric analysis of results from Western blot showing changes in the ratio of dimers versus monomers of eNOS and nNOS protein in RVLM of young and adult rats after saline or l-NAME treatment. Data on protein expression were normalized to the respective saline control value, which is set to 100%. Data are presented as mean ± SD, *n* = 6 per group. * *p* < 0.05 versus corresponding saline-treated group, and ^#^
*p* < 0.05 versus saline-treated young rats in unpaired Student’s *t*-test.

**Figure 6 biomedicines-10-02232-f006:**
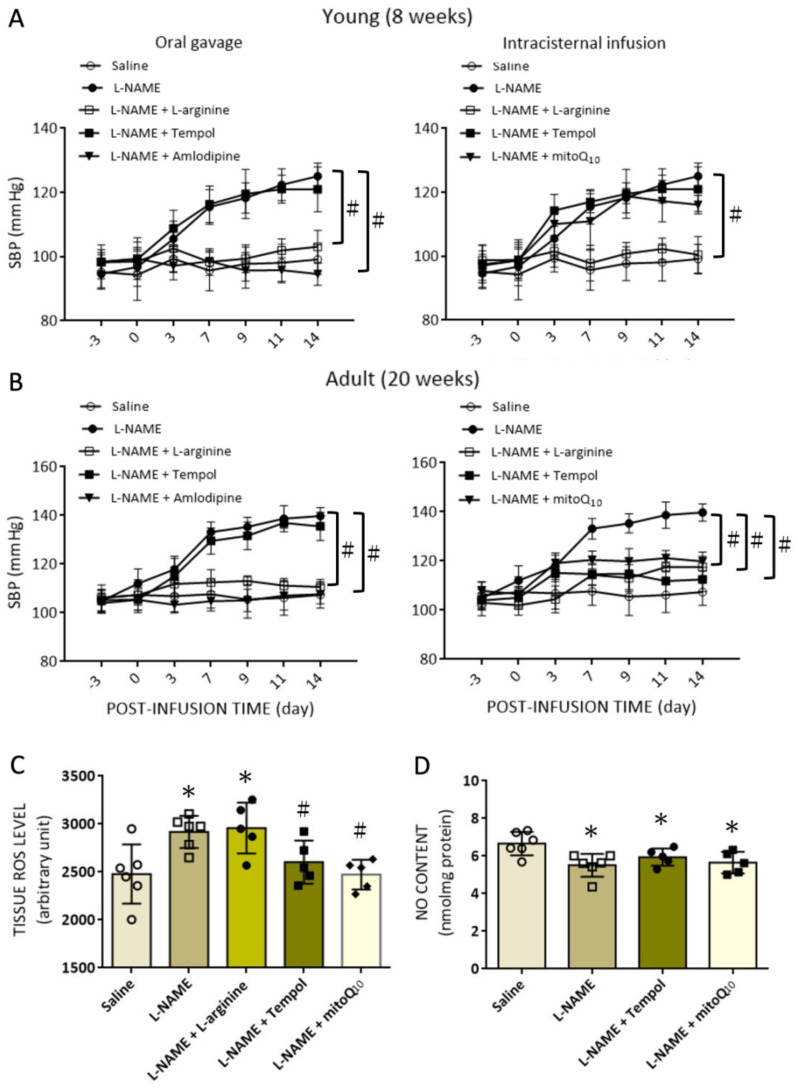
Effect of NO donor and antioxidants on changes in SBP, and tissue ROS and NO levels in RVLM of young and adult rats induced by systemic l-NAME treatment. Temporal changes in SBP at different postinfusion time points after i.p. infusion of l-NAME, alone or with additional oral intake or i.c. infusion of various pharmacological treatments in young (**A**) and (**B**) adult rats. Also shown are tissue levels of ROS (**C**) and NO (**D**) in RVLM, measured at day 14 after i.p. infusion of l-NAME, alone or with additional i.c. infusion of pharmacological treatments. The pharmacological treatments included i.p. infusion of l-NAME (10 mg/kg/day), oral intake via gavage of L-arginine (2%), tempol (100 µmol/kg) or amlodipine (10 mg/kg), or i.c. infusion of L-arginine (2 µg/kg/day), tempol (1 μmol//h/μL), or mitoQ_10_ (2.5 μmol//h/μL). Control infusion of 0.9% saline (for i.p. or oral gavage treatment) or artificial CSF (aCSF; for i.c. infusion) served as the volume and vehicle control. Data are presented as mean ± SD, *n* = 5–6 per group, and * *p* < 0.05 versus the corresponding saline-treated group, and ^#^
*p* < 0.05 versus the l-NAME group in post hoc Tukey’s multiple comparisons tests or unpaired Student’s *t*-test. Data on saline and l-NAME treatments from Figure 2 are adopted for comparison.

**Figure 7 biomedicines-10-02232-f007:**
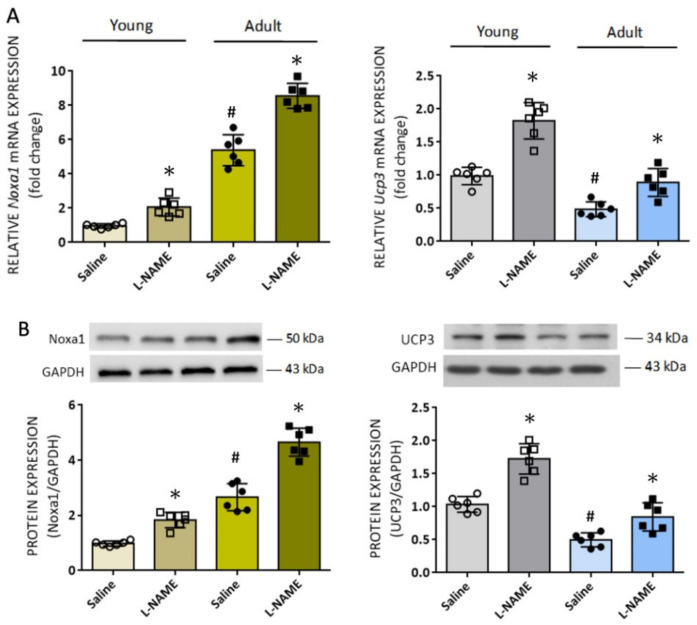
Age-dependent expression of ROS-related proteins in RVLM and the effect of systemic l-NAME treatment on their expressions: (**A**) Relative expression of *Noxa1* and *Ucp3* mRNA quantified by RT-qPCR in RVLM tissues from young (open circules or squares) and adult (filled circles or squares) rats on 14 days after i.p. infusion of saline or l-NAME. (**B**) Representative gels (insets) and densitometric analysis of results from Western blot changes in protein expression of Noxa1 and UCP3 in RVLM of young and adult 14 days after systemic l-NAME treatment. Data on protein expression were normalized to the respective saline control value, which is set to 1.0. Data are presented as mean ± SD, *n* = 6 per group. * *p* < 0.05 versus corresponding saline-treated groups, and ^#^
*p* < 0.05 versus saline-treated young rats in unpaired Student’s *t*-test.

**Figure 8 biomedicines-10-02232-f008:**
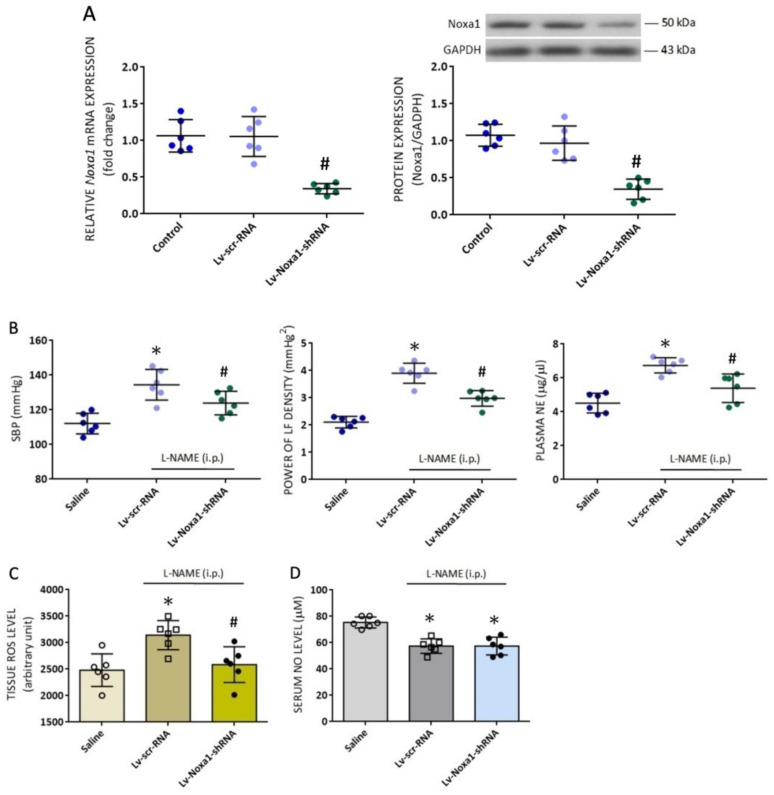
Effect of manipulations of *Noxa1* gene in RVLM on mRNA and protein expression, and changes in cardiovascular responses and levels of ROS in RVLM and serum NO of adult rats induced by systemic l-NAME treatment: (**A**) Changes in mRNA transcription of Noxa1 and representative gels (insets) and densitometric analysis of results from Western blot showing changes in protein expression of Noxa1 in RVLM tissues 4 days after bilateral microinjection into RVLM of lentiviral vectors (Lv) containing short hairpin interfering RNA (shRNA) targeting the rat *Noxa1* sequence (Lv-Noxa1-shRNA) or a scrambled (scr) control shRNA. (**B**) SBP, power density of the LF component of SBP signals, and plasma NE levels; and (**C**) tissue ROS and (**D**) serum NO levels, determined on day 14 after i.p. infusion of saline or l-NAME (10 mg/kg/day) in adult rats that received additional treatment with bilateral microinjection into RVLM of Lv-scr-RNA or Lv-Noxa1-shRNA on day 10 after l-NAME treatment. Data on mRNA transcription and protein expression are normalized to the respective saline control value, which is set to 1.0. Data are presented as mean ± SD, *n* = 6 per group. * *p* < 0.05 versus corresponding saline-treated groups, and ^#^
*p* < 0.05 versus Lv-scr-RNA-treated groups in unpaired Student’s *t*-test.

**Table 1 biomedicines-10-02232-t001:** Changes in plasma levels of proinflammatory cytokines in young (8 weeks) and adult (20 weeks) rats in response to i.p. infusion of saline or l-NAME.

	Saline		l-NAME	
	Young	Adult	Young	Adult
IL-1β (ng/mL)	0.58 ± 0.11	0.47 ± 0.21	0.69 ± 0.31	0.61 ± 0.34
IL-6 (pg/mL)	208 ± 42	186 ± 39	223 ± 68	217 ± 56
TNF-α (pg/mL)	89 ± 38	105 ± 43	93 ± 54	119 ± 72

Saline or l-NAME (10 mg/kg/day) was infused into the peritoneal cavity for 14 days. Data are presented as mean ± SD, *n* = 6 per group. No significant difference exists between groups in One-Way ANOVA. IL-1β, interleukin 1-β; IL-6, interleukin 6; TNF-α, tumor necrosis factor α.

## Data Availability

The data presented in this study are available on reasonable request from the corresponding author.

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
