# Peer review of "Disparate Roles of Oxidative Stress in Rostral Ventrolateral Medulla in Age-Dependent Susceptibility to Hypertension Induced by Systemic l-NAME Treatment in Rats"

_biomedicines, 2022, doi:10.3390/biomedicines10092232_

Round 1

Reviewer 1 Report

The manuscript title “Disparate Roles of Oxidative Stress in Rostral Ventrolateral Medulla in Age-Dependent Susceptibility to Hypertension Induced by Systemic L-NAME Treatment in Rats”. This study explained the role of oxidative stress in RVLM induced by L-NAME in rats different ages. This manuscript is very interesting; however, some points need to be clarified.

1.     Are there any neurological signs after CSF drainage.

2.     Why this study used different methods of NO measurement?

3.     Fifure2A, are there any significant differences of SP between aged 8 and 20 weeks at the beginning of study. The SP of control rats aged 8 weeks is lower than 100 mmHg.?

4.     L-NAME is a non-selective NOS inhibitor, why it reduced only eNOS expression.   

5.     Please explain why the expression p47phox and p67phox subunits did not change in this study.

Minor

1       Introduction line 105-117 should be inserted in results.

2       Please check line 120

3       Please check line 215, intracranial ?

Author Response

Responses to Reviewer #1

We appreciate very much the affirmative views of the Reviewer on our work, and thank you for the opportunity to improve on our manuscript. Please kindly refer to the pdf file of the revised manuscript for page and line number, as they may be changed in Word file with different version of Microsoft Office of your computer.

The manuscript title Disparate Roles of Oxidative Stress in Rostral Ventrolateral Medulla in Age-Dependent Susceptibility to Hypertension Induced by Systemic L-NAME Treatment in Rats. This study explained the role of oxidative stress in RVLM induced by L-NAME in rats at different ages. This manuscript is very interesting; however, some points need to be clarified.

1. Are there any neurological signs after CSF drainage.

Response: We thank the Reviewer for this comment. “Drainage” of CSF from the outer end of the catheter is simply to assure patency of the implanted tubing rather than “draining” all the CSF. There is therefore no observable neurological sign after implantation of the osmotic minipump. To avoid confusion, we have now replaced the word “drainage” with “presence”. The relevant narrative now appears on P. 3, Lines 142-143 and P. 3, Line 149 to P. 4, Line 150.

2. Why this study used different methods of NO measurement?

Response: In this study we used two different methods to measure NO in plasma (colorimetric method) and RVLM tissue (chemiluminescence method). The latter was performed because the small amount of tissue samples collected from RVLM necessitates the need for a more sensitive method to detect NO.

3. Figure2A, are there any significant differences of SP between aged 8 and 20 weeks at the beginning of study. The SP of control rats aged 8 weeks is lower than 100 mmHg?

Response: We added on P. 11, Lines 501-502 that baseline SBP is significantly higher in adult rats (20 weeks) when compared with young rats (8 weeks).

4. L-NAME is a non-selective NOS inhibitor, why it reduced only eNOS expression.

Response: We thank the Reviewer for the comment. Changes in protein expression of nNOS and iNOS isoforms were detected, although they did not reach statistical significance (cf. Figure 3A). This might be related to dose and treatment duration of L-NAME. At a higher dose and longer treatment duration (50 mg/kg/day for 6 weeks), L-NAME results in a decrease in nNOS mRNA in young but an increase in adult rats. The same treatment scheme, on the other hand, has no effect on eNOS mRNA expression in young or adult rats (ref. 24). This was discussed in original manuscript on P. 20, Lines 985-989, and now appears on P. 22, Lines 971-976 in the revised manuscript.

5. Please explain why the expression p47phox and p67phox subunits did not change in this study.

Response: Again, we thank the Reviewer for this comment. We discussed in the revised manuscript (P. 23, Lines 991-992) that no significant change in the expression of p47phox and p67phox subunits in the RVLM following systemic infusion of L-NAME might be related to susceptibility of individual subunit to NO deficiency.

Minor

1. Introduction line 105-117 should be inserted in results.

Response: Per instructions for authors of the journal, we are required to highlight at the end of the introduction the main conclusions of the present study. Nevertheless, we have condensed the text and deleted parts of the narrative (P. 3, Lines 104-116).

2. Please check line 120.

Response: The age of rats indicated on P. 3, Line 120 (now on P. 3, Line 119 in the revised manuscript) is the age of animals when they were purchased. The rats were housed until the age of 6 or 18 weeks before experimental manipulations.

3. Please check line 215, intracranial?

Response: Respectfully we submit that saline was infused directly into the heart via “intracardial” infusion (P. 5, Lines 221-222 in the revision).

Reviewer 2 Report

The study attempted to elucidate the molecular basis of hypertension in an animal model. The research is multifaceted and deepens the understanding of the influence of oxidative stress on the development of this disease. An additional advantage of the study is taking into account the influence of age on the development of these changes.

The work is very well prepared in terms of content and methodology. Nevertheless, it can be improved. Please explain whether the pro-inflammatory cytokines were measured in serum or plasma, as there are inconsistencies in the paragraph describing the assay methods (page 5, lines 203-208). The sampling method for both materials is slightly different.

When giving the device manufacturer, it should provide all the data - device name, manufacturer, place of production. Please check the information on page 5 (line 200), 6 (line 250, line 251, line 263), 7 (line 304).

Statistical analyzes were performed using parametric tests, hence the question was whether the normality of the data distribution was checked, what tests were used?

There is a typo in the text on page 13 (lines 600) and figure 4B (line 621).

Author Response

Responses to Reviewer #2

We appreciate very much the affirmative views of the Reviewer on our work, and thank you for the opportunity to improve on our manuscript. Please kindly refer to the pdf file of the revised manuscript for page and line number, as they may be changed in Word file with different version of Microsoft Office of your computer.

The study attempted to elucidate the molecular basis of hypertension in an animal model. The research is multifaceted and deepens the understanding of the influence of oxidative stress on the development of this disease. An additional advantage of the study is taking into account the influence of age on the development of these changes.

The work is very well prepared in terms of content and methodology. Nevertheless, it can be improved. Please explain whether the pro-inflammatory cytokines were measured in serum or plasma, as there are inconsistencies in the paragraph describing the assay methods (page 5, lines 203-208). The sampling method for both materials is slightly different.

Response: We thank the Reviewer for the comments. The pro-inflammatory cytokines were measured in plasma. The inconsistencies on page 5, lines 203-208 have now been corrected (P. 5, Lines 212-213 in the revised manuscript).

When giving the device manufacturer, it should provide all the data - device name, manufacturer, place of production. Please check the information on page 5 (line 200), 6 (line 250, line 251, line 263), 7 (line 304).

Response: We thank the Reviewer for the reminders. Please note that full information of the device manufacturer was provided in the original manuscript when the device first appeared in text.

  1. 5, line 200; P. 6, line 251, P. 7, line 304 (Now P. 5, Line 205, 212; P. 6, Line 262, 291; P. 7, Line 314): Full information of ThermoFisher Scientific was provided in original manuscript on P. 4, line 191.
  2. 6, line 250: Added Merck KGaA, Darmstadt, Germany on P. 6, Line 260 in the revised manuscript.
  3. 6, line 263 (Now P. 6, Line 273): Full information of Bio-Rad Laboratories was provided in original manuscript on P. 5, line 227.

Statistical analyzes were performed using parametric tests, hence the question was whether the normality of the data distribution was checked, what tests were used?

Response: We thank the Reviewer for this constructive suggestion. Normality of the data distribution was checked before all the statistical analyses using Shapiro–Wilk test to confirm that the data complied with normal distribution (P. 10, Lines 479-481).

There is a typo in the text on page 13 (lines 600) and figure 4B (line 621).

Response: P. 13, line 600 (now P. 14, Line 611) and Figure 4B: MAD has been changed to MDA (P. 15, Line 662).
